# Triflamidation of Allyl-Containing Substances:Unusual Dehydrobromination vs. Intramolecular Heterocyclization

**DOI:** 10.3390/molecules27206910

**Published:** 2022-10-14

**Authors:** Anton S. Ganin, Mikhail Yu. Moskalik, Ivan A. Garagan, Vera V. Astakhova, Bagrat A. Shainyan

**Affiliations:** A.E. Favorsky Irkutsk Institute of Chemistry, Siberian Division of the Russian Academy of Sciences, 1 Favorsky Street, 664033 Irkutsk, Russia

**Keywords:** allyl cyanide, allyl halides, triflamide, oxidative bromotriflamidation, solvent interception, heterocyclization

## Abstract

Allyl halides with triflamide under oxidative conditions form halogen-substituted amidines. Allyl cyanide reacts with triflamide in acetonitrile or THF solutions in the presence of NBS to give the products of bromotriflamidation with a solvent interception, whereas in CH_2_Cl_2_ two regioisomers of the bromotriflamidation product without a solvent interception were obtained. The formed products undergo base-induced dehydrobromination to give linear isomers with the new C=C bond conjugated either with the nitrile group or the amidine moiety or alkoxy group. Under the same conditions, the reaction of allyl alcohol with triflamide gives rise to amidine, which was prepared earlier by the reaction of diallyl formal with triflamide. Unlike their iodo-substituted analogs, bromo-substituted amidines successfully transform into imidazolidines under the action of potassium carbonate.

## 1. Introduction

Oxidative sulfonamidation of unsaturated compounds is a convenient method for the formation of the C–N bond and an expedient route to the synthesis of various linear and cyclic compounds capable of further functionalization. The course of the reaction and the structure of products strongly depend on the reagent, oxidant, and reaction conditions [1,2,3,4]. 

Allylic substrates differ from their vinylic analogs in the possibility of migration of the double bond upon nucleophilic or electrophilic attack of the terminal olefinic carbon atom, which is impossible in vinylic substrates. In the literature, there are not so many examples of the reactions of oxidative sulfonamidation with the participation of allyl-containing substrates. Thus, in the presence of mild oxidant Cu(OAc)_2_ and Cs_2_CO_3_ as a base, N-arylsulfonyl-*ortho*-allylanilines undergo oxidative cyclization to afford the products with four fused rings [5]. Homoallylic aromatic sulfonamides ArSO_2_NHCH(R)CH_2_CH=CH_2_ are intramolecularly oxidized by PhI(OAc)_2_ in the presence of KBr with cyclization to 4-bromopyrrolidines to give a mixture of the *cis* and *trans* isomers in high yield [6]. N-Bromosuccininide (NBS) induced enantioselective cyclization of allyl-N-tosylcarbamates catalyzed by a complex of Sc(OTf)_3_ with chiral phosphine was reported [7]; the yield of the target products, substituted oxazolidinones, reached 71–90%. The latter was easily recyclized to the oxymethyl-substituted aziridines (Figure 1).

Various 6-halomethyl-substituted 1-tosylpiperazin-2-ones were obtained by NBS-induced intramolecular cyclization of N-allyl-N-benzyl-2-(tosylamido)acetamide [8], PdCl_2_(MeCN)_2_-catalyzed cyclization of tosylglycine-N-allylamides with CuCl_2_ in THF [9], or by the combined action of N-chlorosuccinimide (NCS) and PdCl_2_(PhCN)_2_ [10]. The replacement of PdCl_2_(MeCN)_2_ by PdCl_2_(PhCN)_2_ allowed to increase the yield to 90% as compared to 65% in [9] (Figure 2).

Intramolecular bromoamination of *O-*allyl-*N*-hydroxytosylamides via 5-*endo-tet*-cyclization with bromoacetamide proceeds *trans*-diastereoselectively leading to isoxazolidines in good yield and opening a way to aminoalcohols and aziridines as useful building blocks [11] (Figure 3).

Earlier, we studied a lot of unsaturated substrates in the reactions of oxidative sulfonamidation as summarized in review [12] but only a few of them contained electron-withdrawing groups. Among them were divinyl sulfone and divinyl sulfoxide, which reacted with triflamide in the system *t*-BuOCl/NaI via iodotriflamidation with subsequent cyclization into 2,6-diiodo-4-(triflyl)thiomorpholine 1,1-dioxide [13], mono- and diallyltriflamides, which under the same conditions reacted with carboxamides and sulfonamides via halogenation of the double bond [14,15] and/or iodosulfonamidation and cyclization to 3,7-diiodo-1,5-bis(triflyl)-1,5-diazocane and 3,7,9-tris(triflyl)-3,7,9- triazabicyclo[3.3.1]nonane [15], and mono- and diallyl ethers and allyl acetate, which on cooling to −30 °C gave the products of triflamidation or cyclization [16].

## 2. Results and Discussion

With this in mind, in the present work, we have studied the reactions of allyl halides, allyl alcohol, allylamine, acrylonitrile, and allyl cyanide with triflamide under oxidative conditions in different solvents.

The reaction of triflamide (**1**) with allyl chloride (**2**) and allyl bromide (**3**) in the presence of N-bromosuccinimide (NBS) and acetonitrile at room temperature affords the products of halogenation with a solvent interception, *N*-(2-bromo-3-halopropyl)-*N’*- (trifluoromethylsulfonyl)acetamidamides (**4**, **5**) (Figure 4).

Analytically pure compounds were isolated by column chromatography. The structure of compounds **4** and **5** was proved by NMR and IR spectroscopy, as well as elemental analysis data. In particular, the IR spectrum of **4** contains absorption bands at 3334 (*ν*_NH_), 1556 (ν_C=N_), and 663 cm^–1^ (ν_C–Br_). The ^1^H NMR spectrum shows a broadened singlet of the NH group, a triplet of triplets of the CHBr proton, and a singlet at 2.5 ppm, typical for the methyl group in the amidine fragment. The ^13^C NMR spectrum displays the signal of the azomethine group C=N and a quartet of the CF_3_ group. Note, that no products of the addition of the triflamide residue to the double bond were observed.

The reaction of allyl iodide **6** with triflamide under the same conditions gave amidine **5** identical to that obtained in the reaction of allyl bromide **3** (Figure 4). The product does not contain iodine, which is apparently indicative of its substitution in the intermediate bromoiodo derivative **7** by bromine from NBS (Figure 5).

A possible explanation of the formation of the dibromo-substituted amidine **5** from allyl iodide **6** is given in Figure 6, suggesting the bromine/iodine exchange in the intermediate **7**.

Replacing NBS with N-iodosuccinimide (NIS) in the reaction of triflamide with allyl halides **2** and **3**, N-(2-iodo-3-halopropyl)-N’-(trifluoromethylsulfonyl)acetamidamides **8** and **9** were obtained (Figure 7). The low yields in the reaction using NIS can be due to the lower Lewis acidity of the generated iodine cation than that of the bromine cation.

The structure of products **8** and **9** was proved by NMR and IR spectroscopy, as well as elemental analysis data. The IR spectra of both products show two ν_NH_ absorption bands at 3326 and 3231 cm^–1^ and the bands at 1577, 1553 cm^–1^ (ν_C=N_). In the ^1^H NMR spectrum of **8**, a broad singlet of the NH group and a doublet of doublets of the CHI proton appears. The CHI signals in **8** strongly differ from that in **9** in the position and the character of splitting (ddd in **8** and a multiplet in **9**).

Surprisingly, no reaction occurred between allyl iodide **6** and triflamide in the presence of NIS: the reagents were recovered unchanged.

No products could be isolated from the NBS-induced reaction of allyl amine **10** with triflamide because of the strong polymerization of the reaction mixture. In contrast, allyl alcohol **11** afforded a low yield of amidine **12**, which was obtained earlier from the NBS-induced reaction of diallylformal with triflamide (Figure 8) [16].

With acrylonitrile, neither in the system *t*-BuOCl/NaI nor in the presence of NBS, at room temperature or on cooling, any products were isolated, apparently, due to strong polymerization under oxidative conditions (see, e.g., [17]). As distinct from that, the NBS-induced reaction of triflamide with allyl cyanide **13** in acetonitrile gave the product of bromotriflamidation with solvent interception **14** similar to the reactions of other substrates under analogous conditions [16,18]. The yield of N-(2-bromo-3-cyanopropyl)- N’-(trifluoromethylsulfonyl)ethaneimidamide **14** isolated by column chromatography was 60%. Its structure was proved by the methods of IR, NMR spectroscopy, and HRMS. In particular, the IR spectrum of amidine **14** shows absorption bands ν_NH_ (3324), ν_C≡N_ (2259), and ν_NHC=N_ (1560 cm^–1^), its ^1^H NMR spectrum displays a broad NH singlet, the signals of diastereotopic CH_2_N protons and a singlet of the methyl group at the azomethine bond. The ^13^C NMR spectrum contains the C=N and C≡N signals, the CF_3_ quartet, and the corresponding signal appears in the ^19^F NMR spectrum. The use of larger amounts of the reagents allowed to isolate the minor product, N-(2-bromo-3-cyanopropyl)triflamide **15** having no acetonitrile moiety (Figure 9). Its structure was also proved by NMR and IR spectroscopy. The ratio of compounds **14**:**15**, from ^1^H NMR spectroscopy, was ~4:1 (Figure 9).

By replacing acetonitrile with THF as a solvent, we hoped to synthesize amino esters, as was previously successfully completed in our works [18,19]. However, with allyl chloride, instead, the product of bromination, 1,2-dibromo-3-chloropropane **16**, was isolated in a low yield (Figure 10) indicating that triflamide is not involved in the reaction.

The reason for this behavior is that triflamide practically does not react with unsaturated substrates in solvents of low basicity [20].

Carrying out the reaction of allyl cyanide **13** in Figure 9 in THF instead of MeCN also led to the solvent interception product, N-[4-(2-bromo-3-cyanopropoxy)butyl]- triflamide **17** formed via the THF ring opening and its addition as an O-nucleophile (Figure 11).

Excluding the possibility of the formation of amidine **14** by replacing acetonitrile with methylene chloride, we obtained two regioisomers of the product of bromotriflamidation **18** and **19**, isolated them as individual compounds and proved their structure and composition by IR, NMR spectroscopy and elemental analysis. 3-Bromo-4-hydroxybutanenitrile **20** was also obtained in a comparable yield (Figure 12). The prevalence of bromination over bromotriflamidation is probably due to the low solubility of triflamide in methylene chloride.

For comparison, the reaction of allyl cyanide **13** with tosylamide was examined under the same conditions. However, no products of sulfonamidation were obtained, but only dibromide **19** and unreacted tosylamide were recovered.

Amidines **4** and **5** were examined in the reaction with K_2_CO_3_ in acetonitrile. As a result of intramolecular cyclization, substituted 4,5-dihydro-1*H*-imidazoles **21**, **22** were obtained in quantitative yield. However, upon prolonged exposure to humid air, the bromo-substituted imidazoline **22** hydrolyzed to linear adduct **23** (Figure 13):

The structure of imidazolines **21**, **22** was proved by IR and NMR spectroscopy, as well as elemental analysis data. The presence of two NH signals in the ^1^H NMR spectrum, as well as the presence of signals for CH_2_NH, CHNH and C=O groups in the ^13^C spectrum, indicates the formation of adduct **23**.

Amidines **8** and **9** having two halogen atoms could give the products of cyclization with different ring sizes, but neither of them was formed; no reaction with K_2_CO_3_ occurred.

Amidines similar to **14** containing bromine at the β-position to the amidine nitrogen atom readily undergo base-induced intramolecular cyclization to afford 5-substituted 2-methyl-1-triflyl-4,5-dihydro-1*H*-imidazolines in up to quantitative yield [16,19,20,21]. With this in mind, we examined the reaction of amidine **14** with potassium carbonate and triethylamine in acetonitrile and found that dehydrobromination did occur but, instead of the expected 5-cyanomethylimidazoline **24**, N-[3-cyanoprop-2-en-1-yl)]-N’- (triflyl)ethaneimidamide **25** was unexpectedly formed. Even more surprising was the formation of the isomeric N-[3-cyanoprop-1-en-1-yl)]-N’-(triflyl)ethanemidamide **26** in carrying out the two-step reaction of triflamide, alkene, NBS and K_2_CO_3_ using one pot procedure, which was also shown to lead to imidazolines [20], (Figure 14). Replacement of triethylamine or K_2_CO_3_ by sterically hindered 2,4,6-trimethylpyridine (2,4,6-collidine) does not change the course of the reaction under the same conditions, leading to the formation of amidine **25** as the only isomer in 76% yield.

The structure of isomers **25** and **26** was deduced from their ^1^H NMR spectra, in particular, from the multiplicity pattern of the high-field signal of the methylene group. In isomer **25**, the signal of –CH_2_N– group appears as a triplet of doublets at 4.22 ppm due to splitting on the NH and =CH protons with almost equal constants of ~6 Hz, and subsplitting with small constant of 1.5 Hz on the CHC≡N proton. In accordance with this, the CHC≡N signal at 5.65 ppm is detected as a doublet of triplets with coupling constants of 11.2 and 1.5 Hz, and the CH=CHCH_2_ signal at 6.49 as a doublet of triplets with the *J* values of 11.2 and 6.2 Hz. The structure of **25** is unequivocally proved by the 2D ^1^H–^1^H COSY NMR spectrum, which contains cross-peaks between the CH_2_ and NH signals, as well as between the CH_2_ and the signals of the adjacent (more intense) and remote (less intense) vinylic protons (Appendix A Appendix A). The C=C bond is polarized towards the cyano group, Δδ = 0.84 ppm. In contrast, in isomer **26**, the signal of the –CH_2_N– group at 3.28 ppm appears as a doublet of doublets coupled only with the adjacent and remote vinylic protons with *J* = 7.3 and 1.2 Hz, respectively. Both compounds have *trans*-configuration about the double bond. Polarization of the C=C bond in **26** (Δδ = 1.90 ppm) is much larger than in **25**, in compliance with the oppositely directed effects of the CN and –CH_2_N groups in **25**, and the unidirectional effect of the NCCH_2_ and NH groups in **26**.



Earlier, the products of oxidative sulfonamidation with THF interception have been shown to undergo base-induced intramolecular heterocyclization to the corresponding 1,4-oxazocanes [19]. However, as in Figure 15, the reaction of compound **17** with potassium carbonate, instead of cyclization, occurred as dehydrobromination to the isomeric linear products, *N*-(4-((3-cyanoallyl)oxy)butyl)triflamide 27 and *N*-(4-((3-cyanoprop-1-en-1-yl)oxy)butyl)triflamide **28** in the ratio of 1:2 (Figure 15) and the total yield of 80%.

The formation of two regioisomers **27** and **28** by dehydrobromination of ether **17** as distinct from the reaction of amidine **14** (Figure 14) can be due to better conjugation of the C=C bond with the oxygen atom than with the amine nitrogen atom in amidine **26** because of very strong conjugation of the latter in the amidine fragment [22]. The structure of regioisomers **27** and **28** was proved by their ^1^H NMR spectra as described above for regioisomers **25** and **26**.

The proposed pathways for the formation of products **14**, **15**, and **18** are presented in Figure 16. The process could start with the reaction of TfNH_2_ and NBS leading to the reactive species TfNHBr, which acts as a source of electrophilic Br^+^. The latter adds to the double bond of the substrate to give bromonium cation. The further course of the reaction is determined by the reaction medium. In acetonitrile, having higher basicity than triflamide (780 [23] vs. 740 kJ/mol [24]), the molecule of MeCN is captured by the cation with further addition of the triflamide anion to give amidine **14**. A competitive attack of triflamide anion gives rise to a small amount of adduct **15** (Figure 16). In CH_2_Cl_2_, in the absence of an alternative nucleophile, only the isomeric bromamines **15** and **18** are formed via the attack of TfNH¯ on the terminal and internal carbon, respectively, in the intermediate bromonium ion. The formation of dibromide **19** and bromoalcohol **20** (Figure 12) can proceed either by the replacement of the triflamide residue in **15** by bromine or hydroxy group or via the ring opening in the bromonium ion by the terminal attack with these groups.

The most challenging question is why the reaction of dehydrobromination of compound **14** in Figure 15 results in the formation of isomeric linear products **25** and **26**, being drastically different from all earlier studied reactions of similar β-bromoamidines with bases leading to cyclization to imidazolines. The formation of imidazolines in all our previous works is not surprising because of the higher energy of the bonds of different types (C–C, C–N, and C–H in imidazolines vs. C=C and N–H in linear products of dehydrogenation). In the search for a rationale for the specific behavior of compound **14**, we assumed that there could be two reasons for the formation of linear products **25** and **26**: (i) conjugation of the formed C=C bond with the nitrile group in **25** or with the NH group in **26**, and (ii) the presence of acidic NH proton in the amidine motif of **25** and **26**, capable of associating with the basic sites of the second molecule. For this, we performed high-level MP2/6-311++G(d,p) calculations including frequency analysis of molecules **25**, **26**, their dimers, and the isomeric imidazoline **24** shown in Figure 15. The relative energies and free energies are given in Table 1. Remarkably, isomers **25** and **26** form different types of associates: while for **26** it is a 12-membered cyclic dimer with two N–H∙∙∙O=S hydrogen bonds, for the similar dimer of compound **25** the geometry optimization results in its transformation to the eight-membered dimer with two N–H∙∙∙N hydrogen bonds (Figure 1).

The analysis of the data of Table 1 allowed us to explain two apparent inconsistencies with the experiment. First, the Δ*E* and Δ*G* differences of 3–4 kcal/mol between the monomers **25** and **26** seem to contradict the formation of both isomers. However, for the dimers, the corresponding differences in Δ*E* become equal. In spite of different types of H-bonding, the entropy losses upon the formation of dimers in Figure 1 and the Δ*G* values are also equal. Apparently, the lowering of the energy of **25**-dimer is due to higher basicity of the azomethine nitrogen caused by strong conjugation in the NH–C=N tryad, whereas in **26**-dimer this effect is reduced by the rivalry with that in the NH–C=C fragment. Second, while the monomers of amidines **25** and **26** are far less favorable than imidazoline **24**, the dimers are much closer in energy and free energy to this heterocycle. Calculations of higher associates at the used very high level of theory are practically impossible, but the presence of acidic NH protons in monomeric molecules **25** and **26** allows them to be formed. This will certainly further increase the stability of the associates and make it highly probable the reversal of the relative stability with respect to imidazoline **24**.

## 3. Materials and Methods

### 3.1. General Details

All starting materials have been described in the literature. All products were identified using IR, ^1^H, ^13^C, and ^19^F NMR spectroscopy. IR spectra were taken on a Bruker Vertex 70 spectrophotometer in KBr. ^1^H, ^13^C, and ^19^F NMR spectra were recorded in CDCl_3_ or CD_3_CN on Bruker DPX 400 spectrometer at working frequencies 400 (^1^H), 100 (^13^C), and 376 (^19^F) MHz. All shifts are reported in parts per million (ppm) relative to residual CHCl_3_ peak (7.27 and 77.1 ppm, ^1^H and ^13^C), and CFCl_3_ (^19^F). All coupling constants (*J*) are reported in hertz (Hz). Abbreviations are: s, singlet; d, doublet; t, triplet; q, quartet; brs, broad singlet. High-resolution mass spectra were measured on an Agilent 1200 HPLC chromatograph (Palo Alto, CA, USA) with Agilent 6210 mass spectrometer (Santa Clara, CA, USA) (HR-TOF-MS, ESI + ionization in acetonitrile with 0.1% HFBA). Elemental compositions were determined by accurate mass measurement with standard deviation. Melting points were measured on a Boetius apparatus. Flash chromatography was performed using silica gel, 60 Å, 300 mesh. TLC analysis was carried out on aluminum plates coated with silica gel 60 F_254_, 0.2 mm thickness. The plates were visualized using a 254 nm UV lamp.

#### Theoretical Calculations

All structures were optimized without restrictions at the MP2/6-311++G(d,p) level of theory. Frequency calculations were performed on the optimized geometry at the same level of theory. All calculations were performed by the use of Gaussian09 program suite [25].

### 3.2. Synthesis

#### 3.2.1. Reactions of Allyl Halides with Triflamide in the Presence NBS + MeCN

To solution of 1 g (6.7 mmol) of triflamide and 6.7 mmol of allyl halide 1, 2 in 30 mL of acetonitrile added 1.19 g (6.7 mmol) of NBS and reaction mixture was stirred in the dark for 24 h. Solvent was removed in vacuum, then the succinimide was precipitated with diethyl ether, filtered off, and ether removed in a vacuum. Analytically pure samples of substances were separated by column chromatography (0.063–0.2 mm, Acros Organics, Waltham, MA, USA). From the hexane–ether = 1:1 eluate, not reacted triflamide and dibromides were isolated, and from the diethyl ether:hexane = 4:1 or diethyl ether eluates amidines 4, 5 were obtained.

***N-(2-Bromo-3-chloropropyl)-N’-(trifluoromethylsulfonyl)acetamidamide*****, 4.** Yield 0.7 g, 43.2%. Oil. ^1^H NMR (400 MHz, CDCl_3_) δ 6.85 (s, 1H, NH), 4.35 (ddd, *J* = 12.5, 8.3, 4.1 Hz, 1H, CHBr), 4.15 (ddd, *J* = 14.5, 5.9, 4.1 Hz, 1H, CH*^A^*HNH), 3.93 (dd, *J* = 11.8, 4.1 Hz, 1H, CHH*^B^*NH), 3.76 (dd, *J* = 11.9, 8.3 Hz, 1H, CH*^A^*HCl), 3.68 (ddd, *J* = 14.5, 8.3, 5.9 Hz, 1H, CH_2_Cl), 2.53 (s, 3H, CH_3_). ^13^C NMR (100 MHz, CDCl_3_) δ 169.5 (C=NTf), 121.4 (q, *J* = 319.4 Hz, CF_3_), 48.3 (CHBr), 46.6 (CH_2_NH), 45.6 (CH_2_Cl), 22.1 (CH_3_). ^19^F NMR (376 MHz, CDCl_3_) δ −78.99. IR (thin): 3334 (NH), 3137, 2928, 1719, 1566 (C=N), 1430, 1323, 1213, 1198 (CF_3_), 1136, 1086, 1055, 929, 775, 745, 663 (C–Br), 601, 542, 474. HRMS (ESI): *m/z*: [M+H]^+^ calcd for C_6_H_9_BrClF_3_N_2_O_2_S: 343,92087; found [M+H]^+^: 344.92869.

***N-(2,3-Dibromopropyl)-N’-(trifluoromethylsulfonyl)acetamidamide,*****5.** Yeild 1.1 g, 60.1_%_. Oil. ^1^H NMR (400 MHz, CDCl_3_) δ 7.14 (s, 1H, NH), 4.38 (tt, *J* = 8.6, 4.1 Hz, 1H, CHBr), 4.20 (ddd, *J* = 14.6, 6.0, 3.9 Hz, 1H, CH_2_NH), 3.83 (dd, *J* = 10.9, 4.3 Hz, 1H, CH_2_NH), 3.65 (m, 2H, CH_2_Br), 2.52 (s, 3H, CH_3_). ^13^C NMR (100 MHz, CDCl_3_) δ169.7 (C=NTf), 121.4 (q, *J* = 319.6 Гц, CF_3_), 47.7 (CHBr), 47.4 (CH_2_NH), 32.8 (CH_2_Br), 21.9 (CH_3_). IR (thin): 3335, 3227 (NH), 3136, 2943, 1774, 1721, 1580, 1562 (C=N), 1428, 1373, 1323, 1278, 1215, 1197 (CF_3_), 1135, 1081, 1053, 914, 833, 775, 746, 665, 602 (C–Br), 548, 476. ^19^F NMR (376 MHz, CDCl_3_) δ −78.86. Anal. calcd. for (C_6_H_9_Br_2_F_3_N_2_O_2_S): C, 18.48; H, 2.33; F, 14.61; N, 7.18; S, 8.22. Found: C, 18.88; H, 2.60; F, 15.00; N, 7.53; S, 8.56.

#### 3.2.2. Reactions of Allyl Halides with Triflamide in the Presence NIS + MeCN

To the solution of 1 g (6.7 mmol) of triflamide and 6.7 mmol of allyl halide **1**, **2** in 30 mL of acetonitrile added 1.53 g (6.7 mmol) of NIS and reaction mixture was stirred in the dark for 24 h. Solvent was removed in vacuum, then the succinimide was precipitated with diethyl ether, mixture were cooled and succinimide was filtered off, ether removed in a vacuum. Analytically pure samples of substances were separated by column chromatography (0.063–0.2 mm, Acros Organics, Waltham, MA, USA). From the hexane–ether = 1:1 eluate, not reacted triflamide and dibromides were isolated, and from the diethyl ether:hexane = 4:1 or diethyl ether eluates amidines **8**, **9** were obtained.

***N-(2-Iodo-3-chloropropyl)-N’-(trifluoromethylsulfonyl)acetamidamide,* 8.** Yield 0.65 g, 41.4%. Oil. ^1^H NMR (400 MHz, CDCl_3_) δ 6.96 (s, 1H, NH), 4.44 (ddd, *J* = 12.9, 8.6, 4.4 Hz, 1H, CHI), 4.05 (dd, *J* = 9.9, 4.7 Hz, 1H, CH^A^HNH), 4.00 (dd, *J* = 11.7, 4.4 Hz, 1H, CHH^B^NH), 3.81 (dd, *J* = 11.7, 9.9 Hz, 1H, CH^A^HCl), 3.71 (ddd, *J* = 14.4, 8.6, 6.2 Hz, 1H, CHH^B^Cl), 2.52 (s, 3H, CH_3_). ^13^C NMR (100 MHz, CDCl_3_) δ 169.4 (C=NTf), 121.3 (q, *J* = 319.59 Гц, CF_3_), 48.0 (CH_2_NH), 47.5 (CH_2_Cl), 26.1 (CHI), 22.1 (CH_3_). ^19^F NMR (376 MHz, CDCl_3_) δ −78.91. IR (thin): 3326, 3231 (NH), 3080, 2928, 2859, 1723, 1664, 1577, 1553 (C=N), 1428, 1371, 1322, 1215, 1197 (CF_3_), 1140, 1082, 1050, 939, 846, 774, 745, 708, 662 (C–I), 636, 600, 528, 475. Anal. calcd. for (C_6_H_9_ClF_3_IN_2_O_2_S): C, 18.36; H, 2.31; F, 14.52; N, 7.14; S, 8.17; Found: C, 18.50; H, 2.71; F, 15.07; N, 7.53; S, 9.03.

***N-(3-Bromo-2-iodopropyl)-N’-(trifluoromethylsulfonyl)acetamidamide,* 9.** Yield: 0.41 g, 27.5%. Oil. ^1^H NMR (400 MHz, CDCl_3_) δ 7.43 (br.s, 1H, NH), 4.60–4.43 (m, 1H, CHI), 4.14–3.94 (m, 2H, CH_2_NH), 3.78–3.65 (m, 2H, CH_2_Br), 2.53 (s, 3H, CH_3_). ^13^C NMR (100 MHz, CDCl_3_) δ 169.5 (C=NTf), 121.3 (q, *J* = 319.4 Hz, CF_3_), 48.9 (CH_2_NH), 35.1 (CH_2_Br), 25.8 (CHI), 22.1 (CH_3_). ^19^F NMR (376 MHz, CDCl_3_) δ −78.81. IR (thin): 3325, 3241 (NH), 3093, 3036, 2927, 2852, 1727, 1650, 1586, 1555 (C=N), 1428, 1375 (SO_2_), 1321, 1268, 1211, 1195 (CF_3_), 1139, 1081, 1049, 1007, 970, 945, 912, 811, 775, 740, 662, 639, 604, 580, 527, 475. Anal. calcd. for (C_6_H_9_BrF_3_IN_2_O_2_S) C, 16.49; H, 2.08; F, 13.04; N, 6.41; S, 7.34. Found: C, 16.50; H, 2.19; F, 12.73; N, 6.18; S, 7.59.

#### 3.2.3. Reaction of Allyl Alcohol with Triflamide in the NBS + MeCN System

To a solution of 1.00 g (6.7 mmol) of triflamide and 0.39 g (6.7 mmol) of allyl alcohol **4** in 25 mL of acetonitrile was added 1.19 g (6.7 mmol) of NBS, and the reaction mixture was kept in the dark for 24 h. The solvent was removed under reduced pressure, the residue was dissolved in 20 mL of diethyl ether, cooled and the formed succinimide was filtered off. The filtrate was evaporated in vacuum, the residue (1.79 g) was placed on a silica gel column (0.063–0.2 mm, Acros Organics, Waltham, MA, USA) and eluted with ether:hexane = 1:1 mixture, isolating unreacted triflamide (~0.4 g), then with ether, obtaining N-(2-bromo-3-hydroxypropyl)-N’-(trifluoromethylsulfonyl)acetamidamide **12** as a colorless oil.

***N-(2-Bromo-3-hydroxypropyl)-N’-(trifluoromethylsulfonyl)acetamidamide,* 12**. Yield 0.34 g, 26%. The product was obtained earlier and described in [16].

#### 3.2.4. Reaction of Allyl Cyanide with Triflamide in the System NBS+MeCN

To a solution of 1.00 g (6.7 mmol) of triflamide and 0.45 g (6.7 mmol) of allyl cyanide in 40 mL of acetonitrile was added 1.19 g (6.7 mmol) of NBS, and the reaction mixture kept in the dark for 24 h. The solvent was removed under reduced pressure, the residue dissolved in 40 mL of diethyl ether, kept in a refrigerator and the formed succinimide filtered off. The filtrate was evaporated in a vacuum, the residue (~1.81 g) was placed on a silica gel column (0.063–0.2 mm, Acros Organics, Waltham, MA, USA) and eluted with ether:hexane (1:1) to give unreacted triflamide (0.2 g), then with ether to afford 1.08 g N-(2-bromo-3-cyanopropyl)-N’-(triflyl)acetimidamide **14** as a yellow oil.

***N-(2-Bromo-3-cyanopropyl)-N’-(trifluoromethylsulfonyl)acetimidamide***, **14**. Oil. Yield 60%. ^1^H NMR (400 MHz, CDCl_3_) δ 7.21 (br t, *J* = 5.6 Hz, 1H, NH), 4.36 (ddd, *J* = 11.1, 7.1, 5.6 Hz, 1H, CHBr), 3.90 (ddd, *J* = 14.4, 5.6, 5.6 Hz, 1H, CH*^*A*^*N), 3.80 (ddd, *J* = 14.4, 7.1, 5.6 Гц, 1H, CH*^*B*^*N), 3.04 (dd, *J* = 5.6, 2.6 Гц, 2H, CH_2_CN), 2.54 (s, 3H, CH_3_). ^13^C NMR (100 MHz, CDCl_3_) δ 170.2 (C=NTf), 121.4 (q, *J* = 320.0 Hz, CF_3_), 115.9 (C≡N), 48.0 (CH_2_N), 41.8 (CHBr), 25.8 (CH_2_CN), 21.9 (CH_3_). ^19^F NMR (376 MHz, CDCl_3_) δ −78.88. IR (thin): 3324 (NH), 3135, 3025, 2952, 2933, 2259 (C≡N), 1711, 1560 (NHC=N), 1430, 1325, 1195, 1139, 1049, 747, 659 (C–Br), 600, 475. HRMS (ESI): *m/z*: [M+H]^+^ calcd for C_7_H_9_BrF_3_N_3_O_2_S^+^: 335.962919; found: 335.962880.

#### 3.2.5. Reaction of Allyl Cyanide with Triflamide in the System NBS + CH_2_Cl_2_

To a solution of 1.00 g (6.7 mmol) of triflamide and 0.45 g (6.7 mmol) of allyl cyanide in 40 mL of CH_2_Cl_2_ was added 1.19 g (6.7 mmol) of NBS. The reaction was carried out for 24 h in the dark. Then, the solvent was removed under reduced pressure, the residue was dissolved in 40 mL of diethyl ether, placed in a refrigerator for 1 h, and the formed succinimide was filtered off. The ether fraction was evaporated in vacuum, the residue (~2.21 g) was placed on a silica gel column (0.063–0.2 mm, Acros Organics, Waltham, MA, USA) and eluted with hexane to give 3,4-dibromobutanenitrile **19** (0.40 g, 26%), followed by ether:hexane = 1:1, isolating unreacted triflamide (0.6 g), then with ether:hexane (4:1) to afford 3-bromo-4-hydroxybutanenitrile **20** (0.20 g, 18%), and hexane:chloroform:ether (1:2:2) to obtained N-(2-bromo-3-cyanopropyl)triflamide **15** (0.15 g, 19%) and N-(1-bromo-3-cyanoprop-2-yl)triflamide **18** (0.10 g, 13%).

***N-(2-Bromo-3-cyanopropyl)trifluoromethanesulfonamide,* 15.** Yield 19%. Oil. ^1^H NMR (400 MHz, CDCl_3_) δ 6.03 (t, *J* = 5.5 Hz, 1H, NH), 4.18–4.09 (m, 1H, CHBr), 3.67–3.57 (m, 2H, CH_2_N), 2.92 (t, *J* = 6.0 Hz, 2H, CH_2_CN). ^13^C NMR (100 MHz, CDCl_3_) δ 119.3 (q, *J* = 320.1 Hz, CF_3_), 115.3 (C≡N), 49.0 (CH_2_NH), 43.1 (CHBr), 23.4 (CH_2_CN). ^19^F NMR (376 MHz, CDCl_3_) δ −77.23. IR (thin): 3199 (NH), 2923, 2259 (C≡N), 1723, 1615, 1454, 1440, 1380 (SO_2_), 1230 (CF_3_), 1197, 1145, 1098, 1067, 1045, 974, 924, 891, 829, 763, 677, 610, 587, 518. Anal. calcd. for (C_5_H_6_BrF_3_N_2_O_2_S): C, 20.35; H, 2.05; N, 9.49; Br, 27.08; S, 10.87. Found: C, 20.13; H, 2.10; N, 9.98; Br, 27.18; S 10.12.

***N-(1-Bromo-3-cyanopropan-2-yl)trifluoromethanesulfonamide,* 18.** Yield 13%. Oil. ^1^H NMR (400 MHz, CDCl_3_) δ 6.17 (d, *J* = 7.9 Hz, 1H, NH), 4.23 (quint, *J* = 6.3 Hz, 1H, CHNH), 3.71 (t, *J* = 6.0 Hz, 2H, CH_2_Br), 3.09 (d, *J* = 5.7 Hz, 2H, CH_2_CN). ^13^C NMR (100 MHz, CDCl_3_) δ 119.5 (q, *J* = 321.0 Hz, CF_3_), 115.8 (C≡N), 51.8 (CHN), 33.5 (CH_2_Br), 25.1 (CH_2_CN). ^19^F NMR (376 MHz, CDCl_3_) δ −76.8.

***3,4-Dibromobutanenitrile,* 19.** Yield 26%. Oil. ^1^H NMR (400 MHz, CDCl_3_) δ 4.18 (ddd, *J* = 11.8, 6.2, 5.2 Hz, 1H, CHBr), 3.94 (dd, *J* = 11.8, 5.2 Hz, 1H, C*H^A^*HBr), 3.86 (dd, *J* = 11.8, 6.2 Hz, 1H, CH*H^B^*Br), 3.11 (dd, *J* = 17.2, 5.2 Hz, 1H, C*H^A^*HCN), 3.05 (dd, *J* = 17.2, 6.2 Hz, 1H, CH*H^B^*CN). ^13^C NMR (100 MHz, CDCl_3_) δ 116.6 (C≡N), 65.4 (CHBr), 46.1 (CH_2_Br), 24.2 (CH_2_C≡N). IR (thin): 3420, 2957, 2928, 2883, 2257 (C≡N), 2066, 1773, 1723, 1649, 1636, 1625, 1577, 1562, 1546, 1457, 1413, 1379, 1343, 1289, 1198, 1149, 1088, 1060, 1028, 976, 942, 916, 876, 846, 724, 645 (C–Br), 607, 539. Anal. calcd. for C_4_H_5_Br_2_N: C, 21.17; H, 2.22; Br, 70.43; N, 6.17. Found: C, 21.10; H, 2.11.

***3-Bromo-4-hydroxybutanenitrile****,***20.** Yield 18%. Oil. ^1^H NMR (400 MHz, CDCl_3_) δ 4.17 (ddd, *J* = 12.1, 6.0, 5.5 Hz, 1H, CHBr), 3.93 (dd, *J* = 12.1, 5.5 Hz, 1H, C*H^A^*HBr), 3.85 (dd, *J* = 12.1, 6.0 Hz, 1H, CH*H^B^*Br), 3.10 (dd, *J* = 17.3, 5.5 Hz, 1H, C*H^A^*HCN), 3.04 (dd, *J* = 17.3, 6.6 Hz, 1H, CH*H^B^*CN), 2.59 (br.s, 1H, OH). ^13^C NMR (100 MHz, CDCl_3_) δ 116.7 (C≡N), 65.3 (CH_2_OH), 46.0 (CHBr), 24.2 (CH_2_C≡N). IR (thin): 3430, 2962, 2928, 2256, 1783, 1725, 1613, 1543, 1453, 1413, 1380, 1350, 1285, 1228, 1198, 1148, 1087, 1058, 1029, 976, 944, 917, 863, 822, 727, 672, 645, 608, 580, 537, 512. Anal. calcd. for C_4_H_6_BrNO: C, 29.20; H, 3.40; Br, 48.00; N, 9.51. Found: C, 29.29; H, 3.69; Br, 48.72; N, 8.54.

#### 3.2.6. Reaction of Allyl Chloride with Triflamide in the NBS + THF System

To a solution of 1.00 g (6.7 mmol) of triflamide and 0.51 g (6.7 mmol) of allyl chloride **1** in 30 mL of tetrahydrofuran was added 1.19 g (6.7 mmol) of NBS, and the reaction mixture was kept in the dark for 24 h. The solvent was removed under reduced pressure, the residue was dissolved in 20 mL of diethyl ether, mixture was cooled and succinimide was filtered off. The filtrate was evaporated in vacuo, the residue (~2.20 g) was placed on a silica gel column (0.063–0.2 mm, Acros Organics, Waltham, MA, USA) and eluted with ether:hexane = 1:1 mixture, isolating unreacted triflamide, then with ether, obtaining 0.30 g of 1,2- dibromo-3-chloropropane **16** as a yellow oil. Product **16** was obtained and described earlier [26].

#### 3.2.7. Reaction of Allyl Cyanide with Triflamide in the System NBS + THF

To a solution of 1.00 g (6.7 mmol) of triflamide and 0.45 g (6.7 mmol) of allyl cyanide in 40 mL of THF 1.19 g (6.7 mmol) of NBS was added. The reaction was carried out for 24 h in the dark. The solvent was removed under reduced pressure, the residue dissolved in 40 mL of diethyl ether, placed in a refrigerator for 1 h, and the formed succinimide was filtered off. The ether fraction was evaporated in vacuum, and the residue (~2.21 g) was placed on a silica gel column (0.063–0.2 mm, Acros Organics, Waltham, MA, USA) and eluted with ether:hexane (1:1), isolating unreacted triflamide (0.4 g), then with ether:hexane (4:1) to give N-(4-(2-bromo-3-cyanopropoxy)butyl)triflamide **17** (1.16 g, 79%).

***N-(4-(2-Bromo-3-cyanopropoxy)butyl)trifluoromethanesulfonamide****,***17.** (45%). Oil. ^1^H NMR (400 MHz, CD_3_CN) δ 5.86 (s, 1H, NH), 4.24–4.07 (m, 1H, CHBr), 3.85–3.72 (m, 1H, CHBrC*H^A^*HO), 3.72–3.61 (m, 1H, CHBrCH*H^B^*O), 3.61–3.37 (m, 4H), 3.36–3.22 (m, 1H), 3.08–3.00 (m, 1H), 1.79–1.55 (m, 4H). ^13^C NMR (100 MHz, CDCl_3_) δ 119.7 (q, *J* = 321.3 Hz, CF_3_), 116.6 (C≡N), 73.0 (CHBr*C*H_2_O), 70.9 (CH_2_*C*H_2_O), 44.0 (CHBr), 41.8 (CH_2_NH), 27.1 (CHBr), 26.2 (*C*H_2_CH_2_), 26.63 (CH_2_*C*H_2_), 24.6 (*C*H_2_C≡N). ^19^F NMR (376 MHz, CDCl_3_) δ −77.21. IR (thin): 3304, 3221 (NH), 2946, 2876, 2258 (C≡N), 1652, 1452, 1439, 1420, 1373 (SO_2_), 1284, 1230 (CF_3_), 1192, 1148, 1078, 990, 920, 877, 812, 742, 609, 579, 511. Anal. calcd. for C_9_H_14_BrF_3_N_2_O_3_S: C, 29.44; H, 3.84; N, 7.63; Br, 21.76; found: C, 29.90; H, 3.52; N, 7.42; Br, 21.90.

#### 3.2.8. Reaction of N-(2-bromo-3-chloropropyl)-N’-(trifluoromethylsulfonyl)acetamidamide **4** with K_2_CO_3_ in MeCN

To a solution of amidine **4** 0.2 g (0.6 mmol) in acetonitrile (10 mL) was added a 2-fold excess of potassium carbonate 0.17 g (1.2 mmol) and stirred for 4 h. The precipitate in the form of salt was filtered off, the acetonitrile fraction was distilled off under reduced pressure, obtaining 0.13 g of 5-(chloromethyl)-2-methyl-1-((trifluoromethyl)sulfonyl)-4,5- dihydro-1*H*-imidazole **21** as a colorless oil.

***5-(Chloromethyl)-2-methyl-1-(trifluoromethylsulfonyl)-4,5-dihydro-1H-imidazole,*****21.** Yield 0.13 g, 81.3%. ^1^H NMR (400 MHz, CDCl_3_) δ 4.64-4.49 (m, 1H, CHN), 4.07 (d.d.d, *J* = 16.0, 9.3, 1.9 Hz, 1H, CH^A^HN), 3.95 (dd, *J* = 16.0, 1.8 Hz, 1H, CHH^B^N), 3.73 (dd, *J* = 11.5, 6.1 Hz, 1H, CH^A^HCl), 3.67 (dd, *J* = 11.5, 3.1 Hz, 1H, CHH^B^Cl), 2.29 (br. t, *J* = 1.6 Hz, 3H, CH_3_).). ^13^C NMR (100 MHz, CDCl_3_) δ 153.5 (C=N), 118.64 (q, *J* = 322.64 Гц, CF_3_), 61.0 (CHN), 57.5 (CH_2_N), 46.1 (CH_2_Cl), 16.3 (CH_3_). ^19^F NMR (376 MHz, CDCl_3_) δ −74.834. IR (thin): 2947, 2879, 2856, 2622, 1723, 1676, 1438, 1404, 1388 (SO_2_), 1352, 1312, 1283, 1239, 1207 (CF_3_), 1157, 1101, 1075, 1058, 1023, 1003, 939, 898, 872, 784, 767, 736, 681, 666, 621, 592, 536, 505. Anal. calcd. For (C_6_H_8_ClF_3_N_2_O_2_S): C, 27.23; H, 3.05; F, 21.54; N, 10.59; S, 12.11. Found: C, 27.42; H, 3.15; F, 21.67; N, 10.70; S, 12.23.

#### 3.2.9. Reaction of N-(2,3-dibromopropyl)-N’-((trifluoromethyl)sulfonyl)acetamidamide 5 with K_2_CO_3_ in MeCN

To a solution of amidine **5** 0.16 g (0.4 mmol) in acetonitrile (10 mL) was added a 2-fold excess of potassium carbonate 0.11 g (0.8 mmol) and stirred for 4 h. The precipitate in the form of salt was filtered off, the acetonitrile fraction was distilled off under reduced pressure, obtaining 5-(bromomethyl)-2-methyl-1-((trifluoromethyl)sulfonyl)-4,5-dihydro- 1*H*-imidazole **22** as a colorless oil.

**5-(Bromomethyl)-2-methyl-1-(trifluoromethylsulfonyl)-4,5-dihydro-1*H*-imidazole, 22.** Yield 0.11 g, 91.7%. Oil. ^1^H NMR (400 MHz, CDCl_3_) δ 4.59–4.49 (m, 1H, CHN), 4.07 (d.d.d, *J* = 16.0, 9.4, 2.1 Hz, 1H, CH^A^HN), 3.90 (d.d.d, *J* = 16.0, 3.3, 2.1 Hz, 1H, CHH^B^N), 3.57-3.52 (m, 2H, CH_2_Br), 2.27 (br. t, *J* = 1.6 Hz, 3H, CH_3_). ^13^C NMR (100 MHz, CDCl_3_) δ 153.4 (C=N), 121.79 (q, *J* = 324.2 Гц, CF_3_), 60.7 (CHN), 58.5 (CH_2_N), 34.7 (CH_2_Br), 16.4 (CH_3_). ^19^F NMR (376 MHz, CDCl_3_) δ -74.77. IR (thin): 2945, 2878, 2606, 1675, 1437, 1404, 1388 (SO_2_), 1347, 1305, 1238, 1206 (CF_3_), 1156, 1099, 1073, 1054, 1012, 992, 936, 889, 862, 771, 692, 672, 659,641, 613, 590, 574, 535, 477, 416. Anal. calcd. For C_6_H_8_BrF_3_N_2_O_2_S: C, 23.31; H, 2.61; F, 18.44; N, 9.06; S, 10.37. Found: C, 23.42; H, 2.71; F, 18.55; N, 9.13; S, 10.42.

#### 3.2.10. Hydrolysis of 5-(bromomethyl)-2-methyl-1-(trifluoromethylsulfonyl)-4,5-dihydro- 1*H*-imidazole **22**

Compound **17** 0.11 mg (0.36 mmol) was subjected to hydrolysis with the formation of N-(3-bromo-2-((trifluoromethyl)sulfonamido)propyl)acetamide **23**.

***N-(3-Bromo-2-((trifluoromethyl)sulfonamido)propyl)acetamide,* 23**. Yield 0.10 g, 83.3%. ^1^H NMR (400 MHz, CDCl_3_) δ 7.50 (d, *J* = 7.3 Hz, 1H, CHN*H*), 6.33 (t, *J* = 4.8 Hz 1H, CH_2_N*H*), 3.96–3.87 (m, 1H, C*H*NH), 3.69–3.60 (m, 2H, C*H*_2_NH), 3.44 (dd, *J* = 10.9, 7.9 Hz, 2H, C*H*_2_Br), 2.05 (s, 3H). ^13^C NMR (100 MHz, CDCl_3_) δ 173.6 (C=O), 121.67 (q, *J* = 320.94 Гц, CF_3_), 56.0 (CHN), 42.5 (CH_2_N), 32.7 (CH_2_Br), 22.9 (CH_3_). ^19^F NMR (376 MHz, CDCl_3_) δ −77.15. IR (thin): 3352 (NH), 3119 (NH), 2957, 2922, 2853, 2255, 1656 (C=O), 1561, 1547, 1431, 1377 (SO_2_), 1324, 1199 (CF_3_), 1141, 1052, 984, 909, 735, 651, 614, 476. Anal. calcd. For (C_6_H_10_BrF_3_N_2_O_3_S): C, 22.03; H, 3.08; F, 17.42; N, 8.56; S, 9.80. Found: C, 22.19; H, 3.22; F, 17.54; N, 8.63; S, 9.91.

#### 3.2.11. Reaction of N-(2-bromo-3-cyanopropyl)-N’-(triflyl)acetimidamide 14 with Base in Acetonitrile

To a solution of amidine **14** (0.27 g, 0.8 mmol) in acetonitrile (10 mL), 2-fold excess of a base (potassium carbonate or triethylamine) was added and stirred for 2 h. The formed salt was filtered off, the acetonitrile fraction was distilled off in a vacuum, affording N-(3-cyanoallyl)-N’-((trifluoromethyl)sulfonyl)acetimidamide **25** (0.19 g, 93%),

***N-(3-Cyanoallyl)-N’-(trifluoromethyl)sulfonyl)acetimidamide****,***25.** Yield 93%. Oil. ^1^H NMR (400 MHz, CD_3_CN) δ 7.92 (br s, 1H, NH), 6.49 (dt, *J* = 11.2, 6.0 Hz, 1H, =CHCH_2_), 5.63 (dt, *J* = 11.2, 1.4 Hz, 1H, =CHC≡N), 4.20 (td, *J* = 6.0, 1.4 Hz, 2H, CH_2_N), 2.40 (s, 3H, CH_3_). ^13^C NMR (100 MHz, CDCl_3_) δ 171.0 (C=NTf), 148.4 (=CHNH); 121.09 (q, *J* = 319.2 Hz, CF_3_), 116.0 (C≡N), 102.5 (=CHCH_2_), 43.4 (CH_2_); 21.8 (CH_3_). ^19^F NMR (376 MHz, CDCl_3_) δ −79.03. IR (thin): 3323, 3133 (NH), 3082, 2944, 2259 (C≡N), 2228, 1772, 1715, 1661, 1588, 1561, 1427, 1384 (SO_2_), 1354, 1329, 1279, 1216 (CF_3_), 1195, 1141, 1102, 1081, 1068, 1042, 975, 920, 901, 871, 843, 776, 739, 604, 584, 537, 475, 435. HRMS (ESI): *m/z*: [M+H]^+^ calcd for C_7_H_8_F_3_N_3_O_2_S^+^: 256.036757; found: 256.036460.

#### 3.2.12. Reaction of Allyl Cyanide with Triflamide in the System NBS + MeCN + K_2_CO_3_

To a solution of 1.00 g (6.7 mmol) of triflamide and 0.45 g (6.7 mmol) of allyl cyanide in 40 mL of CH_3_CN was added 1.19 g (6.7 mmol) of NBS. The reaction was carried out for 24 h in the dark. Then, 1.85 g (13.4 mmol) of K_2_CO_3_ was added and stirred for another 3 h. The precipitate was filtered off, the solvent removed under reduced pressure, the black residue (~2.43 g) was placed on a silica gel column (0.063-0.2 mm, Acros Organics, Waltham, MA, USA) and eluted with ether:hexane (4:1) giving N-(3-cyanoprop-1-en-1-yl)-N’-(trifluoromethyl- sulfonyl)acetimidamide **26** (1.30 g, 75%).

***N-(3-Cyanoprop-1-en-1-yl)-N’-(trifluoromethylsulfonyl)acetimidamide****,***26.** Yield 75_%_. Oil. ^1^H NMR (400 MHz, CDCl_3_) δ 8.66 (br. s, 1H, NH), 6.94 (t, *J* = 8.8 Hz, 1H, =CHCN), 5.06 (dd, *J* = 16.0, 7.5 Hz, 1H, =C*H*), 3.28 (dd, *J* = 7.5, 1.3 Hz, 2H, CH_2_), 2.56 (s, 3H, CH_3_). ^13^C NMR (100 MHz, CDCl_3_) δ 166.9 (C=NTf), 125.4 (=CHCH_2_), 119.2 (q, *J* = 318.8 Hz, CF_3_); 117.3 (N*C*), 104.5 (=*C*HCN); 29.6 (*C*H_2_NH); 21.6 (CH_3_). ^19^F NMR (376 MHz, CDCl_3_) δ −79.01. IR (thin): 3330, 3120 (NH), 3079, 2958, 2929, 2860, 2256 (C≡N), 2230, 1774, 1711, 1680, 1653, 1576, 1541, 1431, 1381 (SO_2_), 1326, 1267, 1215 (CF_3_), 1194, 1140, 1049, 950, 907, 838, 785, 753, 683, 643, 615, 581, 534, 497.

#### 3.2.13. Reaction of N-(4-(2-bromo-3-cyanopropoxy)butyl)triflamide 17 with a Base in Acetonitrile

To a solution (0.20 g, 0.05 mmol) of N-(4-(2-bromo-3-cyanopropoxy)butyl)triflamide **10** in acetonitrile (10 mL) was added a 2-fold excess of potassium carbonate (0.01 g, 0.1 mmol) and stirred for 4 h. The precipitated salt was filtered off, the acetonitrile fraction was distilled off in a vacuum to afford N-(4-((3-cyanoprop-1-en-1-yl)oxy)butyl)triflamide **27** and N-(4-(3-cyanolyl)oxy)butyl)triflamide **28** in the ratio of 1:2.

*N*-(4-(3-Cyanoprop-1-en-1-yl)oxy)butyl)trifluoromethanesulfonamide, 27; *N*-(4-((3-cyanoallyl)oxy)butyl)trifluoromethanesulfonamide, 28. Oil. ^1^H NMR (400 MHz, CD_3_CN) δ 6.74 (comp. 11; dt, *J* = 16.2, 3.8 Hz, 1H, =C*H*O), 6.57 (comp. 12; dt, *J* = 11.3, 5.6 Hz, 1H, =C*H*CH_2_O), 5.87 (comp. 12; br s, 1H, NH), 5.64 (comp. 11; d, *J* = 16.2, 1H, CH_2_C*H*=CHO), 5.51 (comp. 12; d, *J* = 11.3, 1H, NCC*H*=), 5.45 (comp. 11; br s, 1H, NH), 4.38–4.26 (comp. 12; m, 2H, OC*H*_2_CH_2_), 4.18–4.09 (comp. 11; m, 2H, OC*H*_2_CH_2_), 3.64–3.25 (11 + 12, m), 1.90–1.60 (11 + 12, m). ^13^C NMR (100 MHz, CDCl_3_) δ 150.2, 149.6 (OCH=); 118.6, 118.2 (C≡N); 101.1, 100.2 (=CH); 70.9, 70.7, 69.4, 69.1 (OCH_2_); 44.3, 44.2 (CH_2_NH); 27.57, 27.50, 26.67, 26.50. ^19^F NMR (376 MHz, CDCl_3_) δ −77.18. Anal. calcd. for C_9_H_13_F_3_N_2_O_3_S: C, 37.76; H, 4.58; F, 19.91; N, 9.79; S, 11.20; found: C, 39.02; H, 4.95; F, 21.71; N, 9.02; S, 11.90.

## 4. Conclusions

Substituted amidines were obtained for the first time from allyl halides. Amidines prepared from the reaction of triflamide, allyl halide, NBS, and acetonitrile were successfully converted to the corresponding imidazolidines in good yields. Allyl cyanide reacts with triflamide in the presence of NBS to give, depending on the solvent, different products of oxidative triflamidation. In methylene chloride, two regioisomers of the product of halosulfonamidation are formed, whereas in acetonitrile and THF the main products are those with a solvent interception. The amidine, obtained from the reaction in acetonitrile, behaves differently from all other earlier studied β-bromoamidines, which, when treated with a base, underwent cyclization to imidazolines in quantitative yield. In contrast, N-(2-bromo-3-cyanopropyl)-N’-(triflyl)ethaneimidamide undergoes dehydrobromination with the formation of isomeric linear products, N-[(*E*)-3-cyanopropen-1-yl)]-N’-(triflyl)ethaneimidamides with the new C=C bond in the α- or β-position to the cyano group. In the same manner, N-[4-(2-bromo-3-cyanopropoxy)butyl]triflamide obtained as the solvent interception product from the reaction in THF, was dehydrobrominated to the equimolar isomeric mixture of linear products with the new C=C conjugated with either the cyano group or the oxygen atom. No cyclization occurred to 1,4-oxazocanes as in all other earlier studied similar products. High-level calculations allowed us to explain the observed unusual course of dehydrobromination and the formation of different regioisomers.

## Data Availability

The data presented in this study are available on request from the corresponding author.

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
