# Peer review of "Triflamidation of Allyl-Containing Substances:Unusual Dehydrobromination vs. Intramolecular Heterocyclization"

_molecules, 2022, doi:10.3390/molecules27206910_

Round 1

Reviewer 1 Report

The article is devoted to a relevant  topic with great practical potential. The article is well-written, in accordance with all the requirements of journal. The only small remarks I would note is the grammar and style of the English. It needs to be improved. Overall, I am impressed with the article and believe that it can be accepted in its present form.

Author Response

We can only thank the Reviewer for his so positive evaluation of our work.

Reviewer 2 Report

In the Manuscript ID_molecules-1944736, the interaction of allyl halides, allyl cyanide and allyl alcohol with triflamide in the presence of NBS/NIS was studied. Substituted amidines were obtained with good to moderate preparative yields. On the whole, the work is well done, and the reliability of the experimental data is beyond doubt. In my opinion, this study is of interest to the audience of Molecules.

I would have liked to see more analysis of the results. It seems like this manuscript is missing a section where the reader can generalize the results and analyzed in the bigger picture.

Minor points:

1. Authors are encouraged to proofread the manuscript for typos. For example, p. 3 "Table 6", p. 5 the same, p. 4 "Scheme 10" instead of "Scheme 9", p. 13 the numbers of compounds in the description of the spectra 27 and 28, etc.

2. An analysis of the 1H NMR spectra is not sufficient to establish the structure of isomers 25 and 26. In the opinion of the reviewer, the 13С NMR spectra do not agree with the assignment presented. At a minimum, authors should take advantage of 2D experiments, such as 1H,1H-COSY, 1H,13C-HSQC, 1H,13C-HMBC, 1H,15N-HSQC, 1H,15N-HMBC.

3. What is the reason for the very poor elemental analysis of compound 9?

4. р. 6: 2,4,6-trimethylaniline and 2,4,6-collidine are different substances.

Author Response

1. Authors are encouraged to proofread the manuscript for typos. For example, p. 3 "Table 6", p. 5 the same, p. 4 "Scheme 10" instead of "Scheme 9", p. 13 the numbers of compounds in the description of the spectra 27 and 28, etc.

We thank the Reviewer for attracting our attention to this issue. We tried to do our best in the revision and will do so when reading the proofs. The mentioned typos appeared for no apparent reason only in the pdf file, there is nothing like that in the docx file.

2. An analysis of the 1H NMR spectra is not sufficient to establish the structure of isomers 25 and 26. In the opinion of the reviewer, the 13С NMR spectra do not agree with the assignment presented. At a minimum, authors should take advantage of 2D experiments, such as 1H,1H-COSY, 1H,13C-HSQC, 1H,13C-HMBC, 1H,15N-HSQC, 1H,15N-HMBC.

The Reviewer is absolutely right. We made 2D 1H,1H-COSY experiment, which proved the structure unambiguously, reanalyzed previous spectra, and revised the assignment. The full paragraph about the analysis of the NMR spectra is now rewritten (p. 7).

3. What is the reason for the very poor elemental analysis of compound 9?

We are sorry; erroneusly, the analysis of a crude product before column purification was included. Now corrected.

4. р. 6: 2,4,6-trimethylaniline and 2,4,6-collidine are different substances.

Sorry, of course it was 2,4,6-trimethylpyridine. Corrected.

Reviewer 3 Report

The manuscript submitted by Ganin and coworkers describes the solvent dependent reactivity of triflimide in presence of NBS. The author investigated the reactivity between Allyl halides with triflimide and role of acetonitrile/THF and dichloromethane as a solvent. The author shown the unexpected formed product amidines were successfully converted to the corresponding imidazolidines in good yields. The authors discussed interesting observation of the amidine, obtained from the reaction in acetonitrile, behaves differently from all other earlier studied β-bromoamidines.

Although the paper has many limitations such low yielding reaction, formation of regio-isomer, limited substrate scope and various product formation due to solvent interception. But it will give the interesting idea about reactivity of the triflimide to reader. The work is well-executed and the manuscript written in a clear manner. This referee recommends acceptance of the paper for publication.

Author Response

We thank the Reviewer for his/her positive evaluation of our work and recommendation. We would only like to make two minor remarks about the terminology: (i) we worked with triflamide TfNH2, not triflimide Tf2NH; (ii) formation of various products is not necessarily a drawback, it may be a demonstration of versatile reactivity, which is of special interest.

Round 2

Reviewer 2 Report

The manuscript can be accepted for publication, although the authors have not worked out all the comments of the reviewer.